# MEST: An Action Recognition Network with Motion Encoder and Spatio-Temporal Module

**DOI:** 10.3390/s22176595

**Published:** 2022-09-01

**Authors:** Yi Zhang

**Affiliations:** Department of Computer Science, Sichuan University, Chengdu 610017, China; yi.zhang@scu.edu.cn

**Keywords:** action recognition, temporal modeling, key frame, spatio-temporal information

## Abstract

As a sub-field of video content analysis, action recognition has received extensive attention in recent years, which aims to recognize human actions in videos. Compared with a single image, video has a temporal dimension. Therefore, it is of great significance to extract the spatio-temporal information from videos for action recognition. In this paper, an efficient network to extract spatio-temporal information with relatively low computational load (dubbed MEST) is proposed. Firstly, a motion encoder to capture short-term motion cues between consecutive frames is developed, followed by a channel-wise spatio-temporal module to model long-term feature information. Moreover, the weight standardization method is applied to the convolution layers followed by batch normalization layers to expedite the training process and facilitate convergence. Experiments are conducted on five public datasets of action recognition, Something-Something-V1 and -V2, Jester, UCF101 and HMDB51, where MEST exhibits competitive performance compared to other popular methods. The results demonstrate the effectiveness of our network in terms of accuracy, computational cost and network scales.

## 1. Introduction

Videos contain richer information than single images, including temporal correlations and motion clues between adjacent frames. As a result, temporal modeling becomes a critical step in video action recognition [1]. With the booming development of deep learning, the application of convolution neural networks (CNN) has reached astonishing success in image classification due to their powerful feature learning and reasoning abilities. Albeit effective, it cannot be directly applied to time series signals (e.g., video). To remedy this deficiency, various works have been published to exploit the signal in the temporal dimension, which could roughly be divided into three categories: two-stream architectures, 3D CNN and its variants, and 2D CNN with temporal modules.

As the name implies, a typical two-stream ConvNet architecture has two streams, namely a spatial stream and a temporal stream, wherein the former one distills appearance information from individual RGB frames, and the latter one leverages optical flow as the input to extract motion information. Both streams are implemented based on deep networks. The final result is a fusion of the two streams. It turns out that the appearance and motion information could be effectively integrated by the two-stream architecture [2]. However, the calculation of dense optical flows across adjacent frames in the video sequence is always computationally heavy. An end-to-end action recognition could not be implemented based on a two-stream structure [2]. In this light, 3D CNNs were developed to capture both appearance and temporal information from videos at the same time [3]. Unfortunately, 3D CNNs are extremely difficult to train, due to their large volume of parameters, over-fitting and slow convergence issues, making them difficult to be deployed on normal hardware platforms. In view of this, a lightweight architecture is required to avoid heavy computation. Recently, various works were elaborated by using 2D CNNs as the backbones with additional temporal modeling modules. Du Tran et al. [4] factorized 3D convolution filters into separate spatial and temporal components. Ji et al. [5] shifted part of the channels along the temporal dimension to perform information exchange between adjacent frames without adding extra parameters. Although it reduced the computational load to some extent, it was still weak in temporal information modeling.

Generally, the common problems in existing methods include the following:(1)The optical flow based two-stream architectures suffer high computational complexity.(2)The 3D CNNs based networks have a huge number of network parameters, leading to slow training and convergence, which hinders their applicability.(3)The 2D CNNs based networks model the appearance features of each frame to infer the action type. However, they do not fully explore cross frame motion dynamics, which leads to a weak temporal modeling ability and less satisfactory performance.

Considering the fact that both short-term modeling and long-term aggregation are critical for action recognition [6], a lightweight network is required to model spatial–temporal information with an efficient action recognition algorithm. To this end, MEST is developed in this paper to extract rich short-term and long-term temporal features so as to enhance explicit temporal modelling ability.

Firstly, a temporal shift operation is performed to exchange feature information between consecutive frames. Then a motion encoder is developed to capture motion cues. Next, to reduce computation, the traditional 3D block is factorized into a channel-wise 1D temporal convolution and a 2D spatial convolution to extract spatial–temporal information in videos. With the integration of the three modules, MEST is capable of capturing both distinctive and complementary characteristics of human actions. MEST is compatible with existing CNNs with a relatively small computational cost. Considering the fact that the network scales of the mainstream 2D CNNs for action recognition are still large, causing the problems of slow training and convergence, thus, weight standardization (WS) is applied to the convolution layers to accelerate the training process. To testify the efficacy of MEST, experiments are carried out on five mainstream benchmarks, in which MEST surpasses existing popular methods (e.g., TSM [5] and GST [7]) in terms of accuracy and computational load.

In a nutshell, the main contributions of this paper are summarized as 3-fold:(1)A motion encoder (abbreviated as ME) is presented to capture motion cues across frames without using pixel-level optical flows as additional input.(2)A spatio-temporal module (dubbed SAT) is developed to extract and aggregate long-term spatio-temporal information. The SAT module consists of a channel-wise temporal convolution followed by a 2D spatial convolution, which is used to replace the 3D convolution blocks.(3)The weight standardization (WS) method is applied to the convolution layers in MEST followed by batch normalization (BN) to speed up the training process and convergence.

Comparative results of balanced accuracy and computational complexity on Something-Something-V1 are shown in Figure 1 below (including TSM [5], ECO [8], I3D [9], NL-I3D [10] and MEST). MEST manifests superior performance over other popular methods [5,8,9,10,11] with a relatively small computational load.

The rest of the paper is organized as follows: related works are introduced in Section 2. Our method is described in Section 3. The experimental results with ablation studies are shown in Section 4 with thorough analysis. A final conclusion is drawn in Section 5.

## 2. Related Works

Deep learning has played dominion in the field of action recognition in recent years [12]. Simonyan et al. [2] laid a solid foundation for the two-stream architecture, which trained single RGB frames and dense optical flows separately with CNN to produce a weighted average score. The significance of the two-stream architecture is the exploration of motion information between adjacent frames. To construct long-range temporal features, Wang et al. [11] advised a temporal segment network (TSN) with a sparse sampling strategy to extract short snippets over long videos, which enabled efficient learning through the entire videos without limiting the length of the sequence. Although it achieved great performance, it also relied heavily on pre-computed optical flows.

Another strategy is to use 3D CNN and 3D pooling to extract spatial–temporal features. Wang et al. [13] developed an OF-PCANet method for micro-expression recognition using a spatiotemporal feature learning strategy based on optical flow sequences. Tran et al. [14] learned both appearance and dynamic features via 3D convolution operation (C3D). An improvement was made in this work with an end-to-end architecture. However, the number of parameters involved was extremely large, compared with 2D convolutions. Furthermore, it was difficult to be trained on small datasets such as UCF101, suffering from over-fitting. Under such circumstance, Carreira et al. [9] created a large-scale dataset (Kinetics) to pre-train 3D models. Tran et al. [15] advocated Res3D, which has fewer parameters than C3D with even better results. Huang et al. [16] outlined a differential residual model along with a new loss function to model the movement of eyes. In addition, Qiu et al. [17] declared Pseudo-3D residual networks by decomposing a 3 × 3 × 3 convolution operation into a 1 × 3 × 3 filter and a 3 × 1 × 1 filter, which is applicable to spatio-temporal related tasks. Dong et al. [18] put forward two types of 3D CNNs (residual and attention residual) to improve the performance of the existing 3D CNN.

Although the 3D convolution-based methods obtained good performance, they are computationally heavy. ECO [8] pointed out the major limitations in the current methods, and provided a novel structure using a combination of historical long-term content with a sampling strategy to exploit the spatio-temporal relations between adjacent frames. Zhang et al. [19] eliminated the camera noise in motion using a novel recurrent attention neural network architecture. Zhang et al. [20] mitigated the impact of noisy samples using an auto-augmented Siamese neural network (ASNet). Guo et al. [21] analyzed the multi-modal optimization problems in action recognition and presented a dual-ensemble class imbalance learning method, in which two ensemble learning models were nested with each other. Ji et al. [5] came up with a generic and effective module by shifting part of the channels along the temporal dimensions for information exchange among successive frames. It achieved comparable performance to 3D CNNs while maintaining the complexity of 2D CNNs. GST [7] put forth the idea of grouped convolutions in designing an efficient architecture to separate hierarchical features across the channel. It took advantage of the 3D convolutions to extract features and kept the cost as low as 2D CNNs. Partially motivated by these works, 2D CNNs are used as the backbone of our network. However, unlike most existing methods (that only captured certain kinds of temporal information), both short-term and long-term temporal information are extracted and are combined with motion cues.

As discussed earlier, although successful in action recognition, deep networks are difficult to be trained. To facilitate training and convergence, we utilize batch normalization (BN) [22]. BN regulates certain distributions during training on the basis of data normalization and model initialization so as to avoid degeneracy of the normalization effects. It indeed greatly improves the training process by performing normalization along the batch dimension. Nevertheless, the results of BN largely depend on the batch size. When the batch size decreases, the performance degrades dramatically. For this reason, layer normalization [23] transposed BN into layer normalization by computing the mean and variance from the summed inputs to each neuron in a layer during training. Instance normalization [24] implemented BN for each sample individually. The above normalization methods are all activation based. Instead, weight standardization (WS) [25] is utilized to further elevate the performance of BN. The reason is provided in Section 3.4 along with the implementation details.

## 3. Algorithm Description

In this section, we describe MEST with its core components in detail. The overall structure of MEST is shown in Figure 2 below.

As explained in Section 1, instead of working on a single frame (or stacked frames), the input video is divided into *T* segments $\left\{S_{1},S_{2},\ldots ,S_{T}\right\}$ with equal length. Then, one frame from each segment is randomly selected to produce *T* frames in total via a sparse sampling strategy [11]. An initial prediction of action category can be attained from each snippet in a segment through our network. The video-level action prediction is generated based on a consensus of a series of snippets.

Our network architecture consists of four main components, a temporal shift (TS) module, a spatial and temporal (SAT) module, a motion encoder (ME) and WS method used in each convolution layer, since both spatial and temporal information as well as appearance features are crucial for action recognition. ResNet-50 is chosen as the backbone for two reasons: on one hand, the representation of spatial–temporal information could be enhanced by the TS + ME + SAT structure. On the other hand, the original frame-wise appearance features are also preserved by the residual structure. The 2nd to the 5th layers of the ResNet-50 are all constructed the same way as shown on the right-hand side of Figure 2. First of all, the input features are processed by the TS module (followed by a 1 × 1 Conv layer) to complete the partial information exchange across frames. Then the motion features are captured by the ME module. Next, a SAT module is attached to extract spatial–temporal information. Finally, the WS method is embedded in each convolution layer to expedite training and convergence.

### 3.1. Temporal Shift (TS) Module

TS shifts the feature map along the temporal dimension to realize the information exchange between adjacent frames. The shift operation is illustrated in Figure 3, where a tensor contains *C* channels with *T* frames. The features that are extracted at different times are identified by different colors in each row. Along the temporal axis, we shift 1/8 of the channels by −1, and shift another 1/8 part by +1, while keeping the remaining 3/4 unchanged. TS achieves a comparable temporal modeling ability of 3D CNN, while maintaining the complexity of 2D CNN.

The TS module is attached to the residual branch to reserve the features of the current frame without weakening the spatial learning ability of the original 2D CNN backbone.

### 3.2. Motion Encoder

Motions refer to the movement displacements across frames, reflecting the occurrence of actions. Previous methods depicted motion patterns in the form of optical flow, making the learning of motions independent of the spatio-temporal features [11,13]. Although these methods are proven effective, calculating optical flows from the sequence of images is extremely time consuming.

To alleviate this problem, the motion encoder (ME) is presented, the design intention of which comes from the fact that different channels carry distinct information, some of which tend to model background scenes, while other channels describe dynamic motion patterns. Because of this, it is beneficial to explore motion-sensitive channels.

The architecture of the proposed motion encoder is drawn in Figure 4a. The input feature is represented as a 5D tensor $X^{\prime }\in [N,T,C,H,W]$. We follow the same squeeze and unsqueeze strategy in the SAT module by placing two 1 × 1 2D convolution layers to obtain channel-wise information. After the squeeze operation, we obtain a feature $X_{1}\in \;\left[N,T,C/r,H,W\right]$, where r is a scaling factor (empirically, r = 16 in this module). The motion features at time *t* are represented by the difference between adjacent frames $X_{t-1}$ and $X_{t}$. Instead of subtracting the original features directly, we add a channel-wise transformation on feature vectors to extract motions, which is written as
(1)$$F\left(t\right)=C_{t}*X_{t+1}-X_{t},\;1\leq t\leq T-1$$

Here, $C_{t}$ represents a 3 × 3 2D channel-wise convolution layer, implementing transformation for each channel. *X_t_* refers to the input at time *t*. $F\left(t\right)\in R^{N\times 1\times C/r\times H\times W\;}$ denotes motion feature at time *t*. The motion feature among adjacent frames is concatenated along the temporal dimension. Meanwhile, the motion feature at the last moment is 0 (i.e., $F\left(T\right)=0$). Therefore, the final motion matrix *F* can be written as [$F\left(0\right),\;F\left(1\right),\ldots ,F\left(T-1\right),0$]. Since our main purpose is to find out and excite the motion-sensitive channels, regardless of the detailed spatial layouts. The motion feature *F* is then processed by spatial average pooling as
(2)$$F_{s}=\frac{1}{H\times W}\overset{H}{\underset{u=1}{\sum }}\overset{W}{\underset{v=1}{\sum }}F\left[;,;,;,u,v\right]$$
where $F_{s}\in R^{N\times T\times C/r\times 1\times 1}$, *F*[ ] is the motion matrix. Next, a 1 × 1 2D convolutional layer is applied to expand the channel dimension of the motion feature to the original dimension *C*. The shape of the processed feature $F_{s}^{\prime }$ is [*N, T, C, 1, 1*] and we feed it to a Sigmoid activation function to obtain the mask *M*:(3)$$M=2Sigmoid\left(F_{s}^{\prime }\right)-1,\;M\in R^{N\times T\times C\times 1\times 1}$$

Finally, a residual connection is utilized to preserve the original background information:(4)$$Y^{\prime }=X^{\prime }+X^{\prime }⨀M$$
where *X*′ is the input, *Y*′ is the final output of this module with dimension [*N, T, C, H, W*]. ⨀ stands for a channel-wise multiplication.

### 3.3. (2 + 1)D Spatial and Temporal (SAT) Module

The (2 + 1)D SAT module is designed to learn rich spatial and temporal features by focusing on the main part of action interactions (other than background or other objects). Although stacking 3D convolution blocks in a deep structure is an effective means for temporal modeling, the computational cost also increases exponentially. Instead of using 3D convolution blocks, the SAT module captures both temporal and spatial information at the same time using decoupled 1D + 2D blocks, which simulates the function of 3D convolution with much less computation.

Figure 4b shows the detailed structure of SAT. Suppose that the input feature X is written as a vector $\left[N,T,C,H,W\right]$, where N denotes the batch size, T denotes the temporal dimension, C represents the number of channels, and H and W are the resolution of the input. The channel numbers of the input tensor are squeezed by a scale ratio δ. δ equals the channel number of the input tensor X. This could be written as
(5)$$Y=C_{1}*X$$
where $C_{1}$ is a 2D convolution layer with kernel size 1 × 1. *Y* is the output of the convolution layer, $Y\in R^{N\times T\times 1\times H\times W}$. Next, feature *Y* is reshaped to $Y^{*}\in R^{NHW\times 1\times T}$ for temporal encoding. Since the semantic information of different channels also varied. To be specific, a temporal convolution $C_{2}$ with kernel size 3 is utilized to characterize temporal information for channel-wise features, which could be expressed as
(6)$$Z=C_{2}*Y^{*}$$
where $Z\in R^{NHW\times 1\times T}$, *C*_2_ is a temporal convolution block. Then, we reshape the feature *Z* into $Z^{*}\in R^{N\times T\times 1\times H\times W}$ to model the spatial information using a 2D convolution $C_{3}$ with kernel size 3 × 3 as follows:(7)$$Z^{*}=C_{3}*Z$$
where $Z^{*}\in R^{N\times T\times 1\times H\times W}$, *Z* is the channel-wise features. Finally, a 1 × 1 2D convolution $C_{4}\;$ is utilized to unsqueeze the number of channels to get the output feature $X_{st}\in R^{N,T,C,H,W}$. We implant this operation inside a residual block, and the final output is expressed as
(8)$$X_{st}=C_{4}*Z^{*}+X$$

*Z** represents the spatial information, *C*_4_ is a 1 × 1 2D convolution kernel. In (8), the original frame-level representation and the enhanced spatial-temporal feature is combined via a residual connection. Compared with a standard 3D convolution operation, SAT is computationally more efficient, since the channel dimension of the input feature is reduced by the 1 × 1 convolution and the 3D convolution is decomposed into a 1D temporal convolution followed by a 2D spatial convolution. By inserting the SAT module in ResNet-50, the network is capable of learning long-term spatio-temporal features by focusing on the main part of an action. The encoding of motion cues and the representation of spatio-temporal information are merged into a unified structure through the integration of ME and SAT modules. Different combinations of ME and SAT are studied and analyzed in Section 4.4 below.

### 3.4. Application of Weight Standardization in Convolution Layer

It was revealed that batch normalization (BN) has a huge impact on network training by making the landscape of the optimization problem smoother. As a result, many existing works adopted BN for a faster training and a better convergence solutions. In particular, BN stabilizes the training process by restricting the 1st and 2nd moments of the distribution of outputs in each mini-batch, which is beneficial for training deep structures.

In the meantime, it was observed that BN treats the Lipschitz constants with respect to activation, instead of optimizing the weights directly [26]. In view of this, the weight standardization (WS) method is employed in our network to further smooth the landscape. As shown in Figure 5 below, for each convolution layer (with n filters), a new set of filters is created by using the WS method (i.e., normalized convolution layer), which directly standardizes and optimizes the weights of the convolution layers. Given an input image, a feature map is produced using the new set of filters. Then a batch-normalization (BN) operation is performed to yield the normalized feature map, which is sent to the activation function. Here, the WS method is used to adjust the weights of the filters, and BN is used to process the input data to the activation function. They are deployed in different places and are complementary in expediting convergence.

A standard convolution layer with zero-bias is written as
(9)$$Y_{0}=W*X_{0},\;W\in R^{O\times I}$$
here *X*_0_ and *Y*_0_ denote the input and output features at time 0, respectively. *W* denotes the weights in the convolution layer, while I represents the number of input channels within the kernel region of each output channel, and *O* stands for the number of output channels. Instead of optimizing the loss on the weights directly, WS re-parameterizes the weights W as a function of $W^{\prime }$, and optimizes the loss on $W^{\prime }$ using the stochastic gradient descent (SGD). The relation between W and $W^{\prime }$ is established as
(10)$$W=[W_{i,j}|W_{i,j}=\frac{w_{i,j}^{\prime }-\mu w_{i}^{\prime }}{\sigma w_{i}^{\prime }}]$$
where
(11)$$\mu W_{i,.}^{\prime }=\frac{1}{I}\overset{I}{\underset{j=1}{\sum }}W_{i,j}$$
and
(12)$$\sigma W_{i,.}^{\prime }=\sqrt{\frac{1}{I}\overset{I}{\underset{j=1}{\sum }}W_{i,j}^{2}-\mu^{2}W_{i,.}+\epsilon .}$$

$\mu W_{i,.}^{\prime }$ calculates the mean value of $W_{i,j}$, while $\sigma W_{i,.}^{\prime }$ computes the variance. As a result, WS restricts the 1st and the 2nd moments of the weights of each output channel, respectively, in the convolution layers, and standardizes the weights in a unique way via gradient normalization during back-propagation (BP). Considering that BN normalizes the convolution layers again, we do not implement any affine transformations on W that slow down training. Instead, we only insert WS into convolution layers followed by BN layers during training.

### 3.5. The Unique Features of MEST

The overall architecture is illustrated in Figure 2 in Section 3. We adopt the ImageNet pre-trained ResNet50 as the backbone, followed by the proposed motion encoder and spatial–temporal modules. We utilize the WS method to boost training and convergence.

In general, we have the following distinct points:(1)Existing methods either capture simple motion cues or process appearance features and motion information separately, which results in less satisfactory performance. In comparison, we extract rich frame-wise appearance features and spatial–temporal information based on the designed TS + ME + SAT structure and combine them as a unity.(2)MEST is a 2D structure with limited computational cost, which does not involve any 3D convolutions or optical flow operations.(3)Some of the works adopt batch-normalization method during training, but the performance is affected by the batch size. We make use of the weight standardization (WS) method to realize faster training and better convergence results.

## 4. Experiment and Analysis

In this section, extensive experiments are carried out on five popular public datasets with proven results. First of all, the datasets are introduced with implementation details. Then we compare our results with other popular methods. After that, ablation studies are conducted to verify the effectiveness of each module in MEST. Finally, the comparative and visualization results are created to validate our design.

### 4.1. Datasets

We evaluate the performance of our network on three time-related datasets, Something-Something-V1 and -V2 [27] and Jester [28], and two scene-related datasets, UCF101 [29] and HMDB51 [30]. The number of categories and samples of each dataset are listed in Table 1 below.

Something-Something-V1 [27] is a large-scale labeled video dataset recoding the actions of human in daily life. It consists of 108,499 videos with 174 fine-grained actions. We divide the dataset into training set (86,017 videos), validation set (11,522 videos) and test set (10,960 videos) following the official guideline. The general performance is reported on the validation set.

Something-Something-V2 [27] is the 2nd release of V1 with updates in four aspects: (1) The number of videos is expanded to 220,847; (2) V2 provides object annotations in training and validation sets. For example, V2 annotates the action as “Putting an apple on the table”, instead of “Putting [something] on [something]” as appears in V1. In total, there are 30,408 objects with 318,572 annotations; (3) The crowd-sourcing method is used in V2 to test the video quality and verify the correct answers for each video; (4) The height for each video is increased to 240 px (it is only 100 px in V1).

Jester [28] is a third-person view gesture dataset, which has a potential usage for human computer interaction. It has 27 categories with 118,562 training videos, 14,787 validation videos and 14,743 testing videos.

UCF101 [29] is a classic dataset of action recognition. It contains 13,320 video sequences from YouTube, with a total of 101 categories.

HMDB51 [30] contains 6766 videos with a total of 51 action categories. Most of the videos in HMDB51 are taken in real scenes, including a large number of facial and limb movements.

Both UCF101 and HMDB51 are scene-related datasets, wherein most of the actions can be reasoned by the background information. For instance, brushing teeth and making up. However, a strong temporal modeling ability is required to recognize the actions in the time-related datasets (Something-Something-V1 and -V2 and Jester). Some symmetrical actions cannot be identified merely based on an individual frame (e.g., “pushing something from left to the right” vs. “pushing something from right to the left”).

In Figure 6, the label for the first row is “closing the dishwasher”. However, if we reverse the order of the frames, the action changes to “opening the dishwasher”. Therefore, the recognition results of the time-related datasets strongly reflect the temporal modeling ability of our model and demonstrate the efficiency of our method. Our research mainly focuses on these time-related datasets, which is also effective for scene-related datasets.

### 4.2. Implementation Details

We utilize ResNet-50 as the backbone in our experiments and sample 8 or 16 frames from each video following the sparse sampling strategy. The length of the short side of the frames is fixed as 256. Both corner cropping and random scaling are employed for data augmentation during training, and finally, each cropped region is resized to a patch of 224 × 224. For Something-Something-V1 and -V2, we pre-train our model on ImageNet. The batch size is 22, and the initial learning rate is set to 0.01 (60 epochs in total, and decays 0.1 at 20, 40 and 50 epochs). We train our network using the SGD algorithm with weight decay 1 × 10^−4^ and dropout 0.5. For Jester, the batch size is 22, and the initial learning rate is set to 0.01(30 epochs in total, and decays by 0.1 at 10, 20 and 25 epochs) with weight decay 1 × 10^−4^ and dropout 0.5. For UCF101 and HMDB51, we pre-train our model on Kinetics-400. The batch size is 22 and the initial learning rate is 0.001 (30 epochs in total, and decays by 0.1 at 10, 20, and 25 epochs) with weight decay 1 × 10^−4^. Considering that UCF101 and HMDB51 are small-scale datasets, the dropout rate is set to 0.8 to prevent over-fitting. We train our model on a NVIDIA RTX 3090 GPU.

Theoretically, accuracy and computational cost are the main concerns for an action recognition task. However, different application scenarios have different focus. When accuracy is the primary concern, we follow the configuration suggested by [10] and sample two clips per video and full resolution input with shorter side 256 for evaluation. Reversely, when the computational cost becomes the main concern, we use only one clip per video and use center 224 × 224 patch for evaluation.

### 4.3. Experiment Results and Comparison with the State-of-the-Art Methods

In this section, we compare the performance of our network with TSN [11] and TSM [5] on all three time-related datasets (Something-Something-V1 and -V2 and Jester). For a fair comparison, they all use eight frames as input and sample 1 clip per video and use center 224 × 224 crops for evaluations. Moreover, we also compare MEST with TSN and TSM on scene-related datasets (UCF101 and HMDB51) using 16 frames as the input.

As shown in Table 2 below, TSN makes poor results due to the lack of the ability of temporal information modeling. TSM extracts temporal information from sequential signal, but it is weak in explicit temporal modeling, while our network exceeds them in both spatiotemporal and motion modeling by a large margin. Compared with TSM, we improved the accuracy of Top-1 by 2.2% on Something-Something-V1, 1.0% on Something-Something-V2 and 0.9% on Jester, respectively. As for scene-related datasets, compared with TSM, we also improved the Top-1 accuracy by 0.9% on UCF101 and 2.7% on HMDB51.

Next, we compare our method with the state-of-the-art methods on five time-related and scene-related datasets. There are five metrics used to measure the combined performance: accuracy for Top-1 and Top-5, number of frames required, FLOPS and network scales. The number of frames by each method is reported with corresponding computational loads in the form of FLOPS and model scale. As expected, sampling more frames will undoubtedly increase the accuracy, which would also inevitably increase the FLOPS at the same time.

Table 3 summarizes the results on Something-Something-V1, including the results by using 2D CNN methods and 3D CNN methods. We report the performance of our method for both 8-frame input and 16-frame input and the ensemble results. First of all, we compare MEST with 2D-CNN-based baselines [13,31] in terms of late fusion for long-range temporal modeling. The comparative results prove that MEST goes far beyond the baseline method on Something-Something-V1. Then, we compare with some recent 3D-CNN-based methods, including I3D [9] and non-local I3D [10]. Apparently, MEST yields better results than 3D-CNN-based models with much lower computational cost and a smaller model size. Moreover, MEST achieves much higher accuracy than non-local I3D [10] with far less FLOPs on the validation set. Finally, we compare with other methods with temporal modules (e.g., TSM [5] and TANet [32]), and MEST again outshines them in terms of accuracy and computational complexity, which validates the effectiveness of our design. ECO [8] used early 2D + late 3D architecture to realize medium-level temporal fusion. MEST outperforms ECO by a large margin with a much smaller model size. For instance, MEST achieves 47.8% Top-1 accuracy using only 8 frames as input, which is still 1.4% higher than that of ECO with 92 frames as input. TSM [5], TANet [32] and SmallBigNet [33] obtain 49.7%, 50.6% and 50.4% Top-1 accuracy, respectively when they combine the results of the 8-frame input and 16-frame input, while MEST obtains a better performance of 52.8% Top-1 accuracy. Table 4 summarizes the results on Something-Something-V2 in comparison with state-of-the-art methods. By using only eight frames as input, our model produces 60.1% Top-1 accuracy, which again outshines other methods which sample eight frames or more than eight frames as input. When using 16 frames as input, we increase the accuracy to 61.3%. Finally, we use the ensemble of 8-frame input and 16-frame input and achieve 64.1% Top-1 accuracy. Table 5 shows the performance on Jester. MEST obtains 96.6% Top-1 accuracy with 8 frames input by using 2 clips per video for evaluation.

Finally, we report the comparison results on scene-related datasets: HMDB51 and UCF101 in Table 6. MEST achieves 96.8% and 73.4% Top-1 accuracy on UCF-101 and HMDB51, respectively. These two datasets are relatively small, suffering over fitting. So, we pre-train MEST on Kinetics-400 and migrate the learned models to UCF101 and HMDB51. We compare MEST with previous state-of-the-art methods, such as 2D baselines of TSN, 3D CNNs of C3D and P3D and other temporal modeling methods [5,31], where MEST manifests impressive results. The reason behind these results is 3-fold: firstly, MEST extracts diverse and abundant spatial–temporal information compared to existing methods, which merely capture appearance features; secondly, MEST explores motion-sensitive channels to describe motion patterns; thirdly, MEST merges the above two pieces of information into a unified framework to achieve better performance than the previous methods.

### 4.4. Ablation Study

In this section, we perform several ablation studies on the Something-Something-V1 and Jester datasets to verify the effectiveness of each module in MEST. We apply the 1-clip and center crop testing method for evaluation and report the Top-1 result, where eight frames are sampled from each video of the training set as input to the network.

Investigate the functions of the two modules and WS method:

To validate the contributions of each component (SAT, ME module and WS method) in our network, the results from the individual modules and combination of modules are listed and compared in Table 7. The contribution of each module to the overall performance is quite obvious. Here, we choose ResNet-50 with the temporal-shift module as the baseline. Our method achieves better performance than the baseline method. Specifically, with only the SAT module, we gain an accuracy of 45.9% Top-1 on Something-Something-V1 and 95.1% on Jester, but it brings in 0.19 M more parameters; with only ME module, we achieve an accuracy of 46.0% on Something-Something-V1 and 95.6% on Jester with an additional 0.85 M parameters. With ME + SAT, we increase the accuracy to 46.6% and 96.2% with 25.73 M parameters. Finally, with the WS method, the accuracy is further increased to 47.8% and 96.6%, respectively, which verifies its functionality.

Investigate different ways of integrating the two modules:

In this section, ME and SAT are deployed in both parallel and serial ways (Mode 1 to 4) to test their combined effectiveness:(1)Firstly, the SAT module and ME module are integrated via element-wise addition operation and are deployed after the first Conv1 × 1 of each bottleneck layer (shown in Figure 7a);(2)Secondly, the SAT module and ME module are integrated via element-wise addition, but we append them after Conv 3 × 3 (shown in Figure 7b);(3)Thirdly, the SAT module is placed in each bottleneck layer (after the 1st Conv1 × 1) and appends the ME module after the 3 × 3 Conv layer of all bottleneck layer (shown in Figure 7c);(4)Finally, the positions of SAT and ME are swapped on the basis of mode 3 to create mode 4 (shown in Figure 7d).

Experimental results of the above four modes are listed and compared in Table 8 below. It turns out that we obtain better performance by connecting them in a serial way. In particular, by placing ME before SAT, we can achieve the highest Top-1 accuracy. We believe the reason is that ME calculates the short-term frame-wise motion cues, which are more suitable to be placed at the early stage of the network, while SAT excels in long-term temporal modeling, which is more suitable for late-stage processing.

Testifying the impact of WS in facilitating training and convergence:

We propose the WS method (working together with BN) to accelerate the training process and expedite convergence. In this section, we use Something-Something-V1 as the dataset by sampling eight frames as the network input and compare the loss values over iteration during training between the network with WS and without WS so as to demonstrate the function of WS method. The results are shown in Figure 8 below. The blue curve characterizes loss values without WS, while the orange curve indicates the values with WS. Obviously, the orange curve decreases faster than the blue curve, and its value at the 55th epoch is lower than that of the blue curve. Since the loss value directly demonstrates the speed of training, we thereby conclude that WS indeed facilitates training and convergence.

### 4.5. Confusion Matrix of the Proposed Method

Confusion matrix is a standard format for accuracy evaluation. Figure 9 shows the confusion matrices created by MEST on Jester.

Each column of the confusion matrix represents the predicted label of the video, and each row represents the real label of the video. Since Jester has 27 categories, we represent these 27 categories using numbers 1~27. For example, the number 21 stands for “thumbs up” and the number 22 stands for “thumbs down”. The value on the diagonal of the matrix represents the proportion of video samples that is correctly classified. Therefore, the more that predicted categories fall on the diagonal, the better the recognition performance. Apparently, almost all categories on Jester are correctly classified, which indicates the strong discrimination ability of MEST.

### 4.6. Visualization of Activation Maps

Figure 10 shows the visualization results on Something-Something-V1 produced by GradCAM [36].

For the sake of simplicity, we merely generate the activation maps in the center frames, taking eight frames as input of the network. The left column shows the raw frames sampled from videos. The middle column indicates the activation results of TSM. The right column displays our results. As shown in Figure 10, the activation maps reflect the fact that the TSM (baseline) simply focuses on the objects, while MEST precisely focuses on the motion-related region and interaction between human hands and objects, owing to the strong temporal modeling ability of the TS + ME + SAT structure.

## 5. Conclusions

Aiming at the extraction of both spatial–temporal and appearance information to create an efficient action recognition network, we propose MEST in this paper. Firstly, a (2 + 1)D spatial and temporal module (called SAT) is proposed by factorizing the 3D block into the 1D + 2D structure to circumvent heavy computation. Then, a motion encoder (ME) is employed to capture the motion cues, which is then integrated with spatial–temporal information into a unified framework. Finally, apart from BN that some of previous methods used for the training process, the WS method is utilized to further boost the training and convergence results. Our network is simple yet efficient, and does not involve any 3D blocks or optical flow operations. Extensive experiments were conducted on five mainstream datasets to compare the overall performance of our approach compared to other popular methods. The reported results validate the efficacy of our network in reaching a high accuracy while maintaining low computational costs.

Nevertheless, there are some shortcomings that need to be solved: Firstly, the sparse sampling algorithms could be improved. Instead of randomly selecting 1 frame from each sequence, we should highlight the frames with changing motions. Additionally, despite no 3D blocks being used, our network structure itself is still riddled with redundancies. In the future, we are committed to network pruning to create more lightweight models for action recognition.

## Figures and Tables

**Figure 1 sensors-22-06595-f001:**
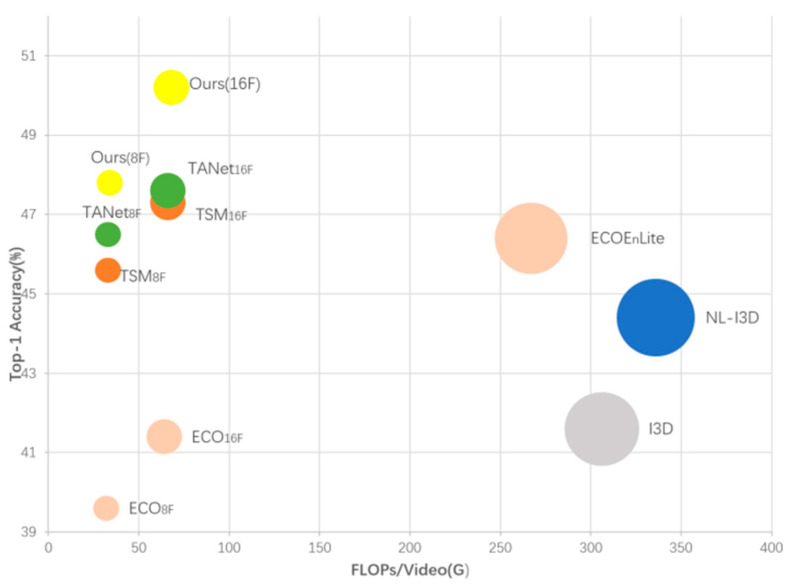
A comparison of performances in terms of Top 1 vs. computational cost on Something-Something V1 (note that the larger the circle the higher the computational cost).

**Figure 2 sensors-22-06595-f002:**
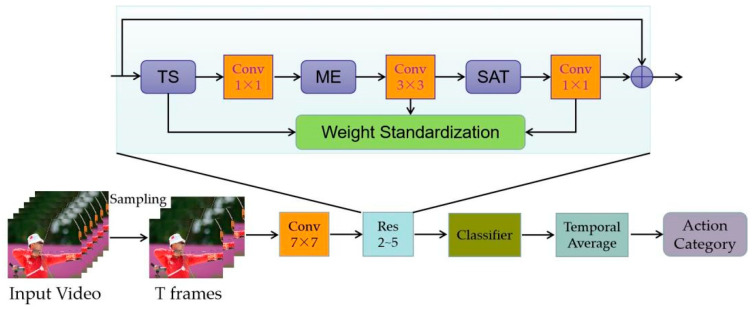
The overall structure of MEST.

**Figure 3 sensors-22-06595-f003:**
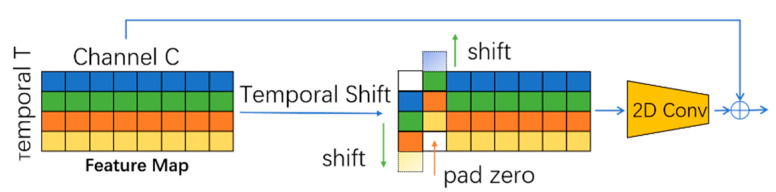
The structure of a temporal shift (TS) module.

**Figure 4 sensors-22-06595-f004:**
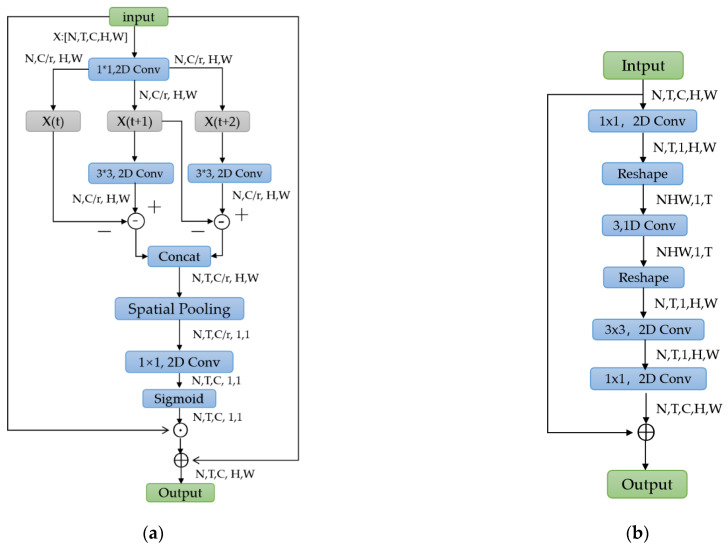
The structures of the proposed motion encoder (ME) and (2 + 1)D spatial and temporal module. (**a**) The structure of motion encoder; (**b**) The structure of (2 + 1)D spatial and temporal module.

**Figure 5 sensors-22-06595-f005:**
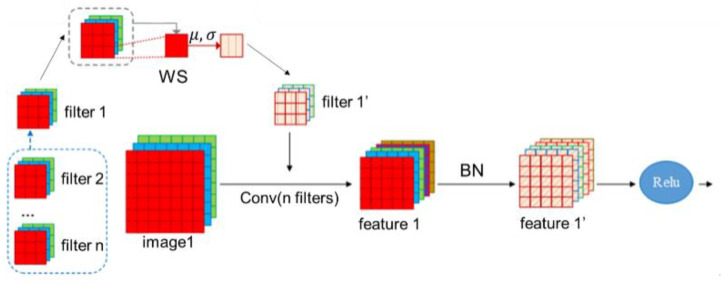
Process of weight standardization.

**Figure 6 sensors-22-06595-f006:**
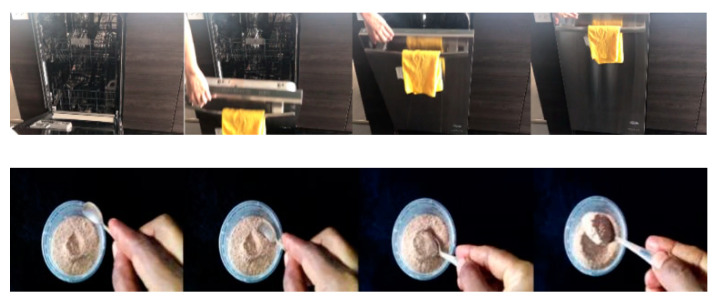
Example of actions in time-related datasets (**top**) closing the dishwasher; (**bottom**) scooping a spoonful of powder).

**Figure 7 sensors-22-06595-f007:**
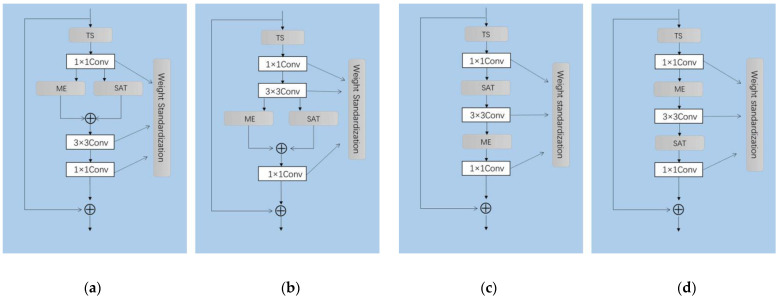
Different layouts of integrating the proposed 2 modules. (**a**) Mode 1; (**b**) Mode 2; (**c**) Mode 3; (**d**) Mode 4.

**Figure 8 sensors-22-06595-f008:**
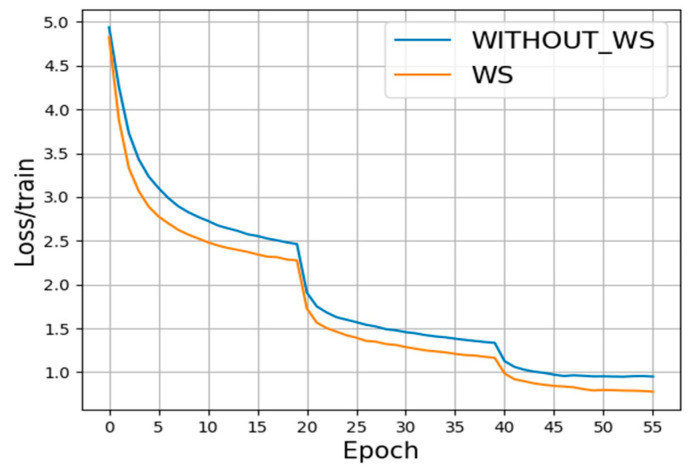
The values of loss over iteration during training process.

**Figure 9 sensors-22-06595-f009:**
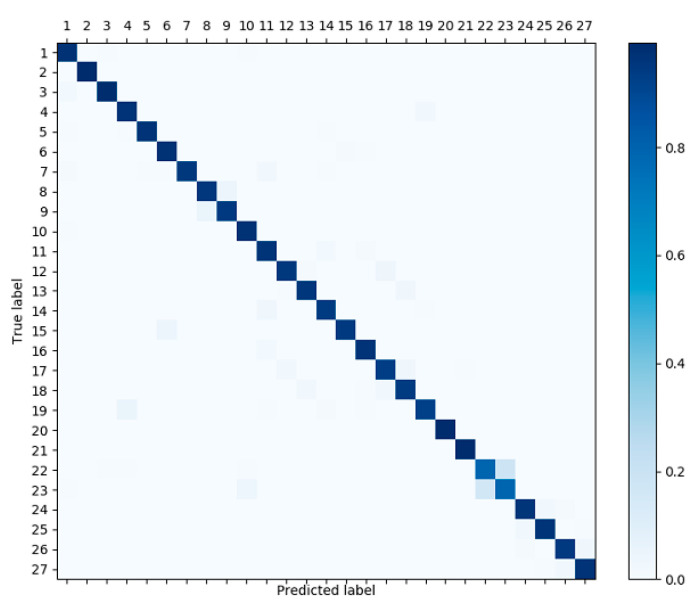
Confusion matrix created by MEST.

**Figure 10 sensors-22-06595-f010:**
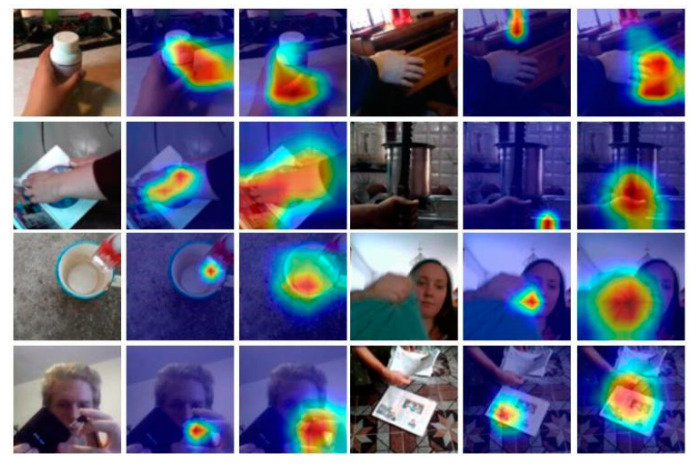
Visualization of activation maps. The first and the 4th columns refer to the original video; The second and the 5th column refer to the result of the baseline; The 3rd and the 6th column refer to our results.

**Table 1 sensors-22-06595-t001:** Public datasets of human action recognition.

Dataset	Category	Samples
Something-Something-V1	174	108,499
Something-Something-V2	174	220,847
Jester	27	148,092
UCF101	101	13,320
HMDB51	51	6766

**Table 2 sensors-22-06595-t002:** Comparison with baseline on three benchmarks.

Dataset	Model	TOP-1%	ΔTOP-1%
Something-Something-V1	TSN	19.5	-
	TSM	45.6	+26.1
	Ours	47.8	+28.3
Something-Something-V2	TSN	27.8	-
	TSM	59.1	+31.3
	Ours	60.1	+32.3
Jester	TSN	81.0	-
	TSM	94.4	+13.4
	Ours	95.3	+14.3
UCF101	TSN	94.0	-
	TSM	95.9	+1.9
	Ours	96.8	+2.8
HMDB51	TSN	68.5	-
	TSM	70.7	+2.2
	Ours	73.4	+4.9

**Table 3 sensors-22-06595-t003:** Results on Something-Something-V1 compared with state-of-the-art methods (center crop, 1 clip).

Model	Backbone	# Frame	FLOPs/Video	# Param	Top-1%	Top-5%
TSN-RGB [13]	BNInception	8	16 G	10.7 M	19.5	-
TRN-Multiscale [31]	BNInception	8	33 G	18.3 M	34.4	-
Two-stream TRN [31]	BNInception	8 + 8	-	36.6 M	42.0	-
I3D [11]	3D ResNet-50	32 × 2	153 G × 2	28.0 M	41.6	72.2
Non-local I3D [10]	3D ResNet-50	32 × 2	168 G × 2	35.3 M	44.4	76.0
Non-local I3D + GCN [10]	3D ResNet-50 + GCN	32 × 2	303 G × 2	62.2 M	46.1	76.8
ECO [8]	BNIncep + 3D Res 18	8	32 G	47.5 M	39.6	-
ECO [8]	BNIncep + 3D Res 18	16	64 G	47.5 M	41.4	-
ECOEnLite [8]	BNIncep + 3D Res 18	92	267 G	150 M	46.4	-
ECOEnLiteRGB + Flow [8]	BNIncep + 3D Res 18	92 + 92	-	300 M	49.5	-
TSM [5]	ResNet50	8	33 G	24.3 M	45.6	74.2
TSM [5]	ResNet50	16	65 G	24.3 M	47.2	77.1
TSMEn [5]	ResNet50	8 + 16	98 G	48.6 M	49.7	78.5
TANet [32]	ResNet-50	8	33 G	25.6 M	46.5	75.8
TANet [32]	ResNet-50	16	66 G	25.6 M	47.6	77.7
TANetEn [32]	ResNet-50	8 + 16	99 G	51.2 M	50.6	79.3
SmallBig [33]	ResNet-50	8	52 G	-	47.0	77.1
SmallBig [33]	ResNet-50	16	105 G	-	49.3	79.5
SmallBigEn [33]	ResNet-50	8 + 16	157 G	-	50.4	80.5
GST [7]	ResNet-50	8	29.5 G	-	47.0	76.1
GST [7]	ResNet-50	8 × 2	29.5 G × 2	-	47.6	76.6
GST [7]	ResNet-50	16	59	-	48.6	77.9
Ours	ResNet-50	8	34 G	25.7 M	**47.8**	**77.1**
Ours	ResNet-50	16	67 G	25.7 M	**50.1**	**79.1**
Ours	ResNet-50	8 + 16	101 G	51.4 M	**52.8**	**81.4**

**Table 4 sensors-22-06595-t004:** Comparison of performance on Something-Something-V2.

Model	Backbone	# Frame	FLOPs/Video	Pre-Train	Top-1%	Top-5%
TSN-RGB [13]	BNInception	16	32 G	Kinetics	30.0	60.5
TRN-Multiscale [31]	BNInception	8	33 G	ImageNet	48.8	77.6
Two-stream TRN [31]	BNInception	8 + 8	-	-	55.5	83.1
TSM [5]	ResNet50	8	33 G	Kinetics	59.1	85.6
CPNet [34]	ResNet-50	24	-	ImageNet	58.7	84.8
Ours	ResNet-50	8	34 G	ImageNet	**60.1**	**86.4**
Ours	ResNet-50	16	67 G	ImageNet	**61.3**	**87.7**
Ours	ResNet-50	8 + 16	101 G	ImageNet	**64.1**	**89.3**

**Table 5 sensors-22-06595-t005:** Comparison of performances on Jester.

Model	Backbone	# Frame	FLOPs/Video	Pre-Train	Top-1%	Top-5%
TSN-RGB [13]	BNInception	8	16 G	Kinetics	81.0	99.0
TSN-RGB [13]	BNInception	16	32 G	Kinetics	82.3	99.2
TSM [5]	ResNet50	8	33 G	Kinetics	94.4	99.7
TSM [5]	ResNet50	16	65 G	Kinetics	95.3	99.8
Ours	ResNet-50	8	34 G	ImageNet	95.3	99.7
Ours	ResNet-50	8 × 2	34 G × 2	ImageNet	**96.6**	**99.9**

**Table 6 sensors-22-06595-t006:** Comparison of performances on UCF101 and HMDB51.

Model	Pre-Train	Backbone	UCF101	HMDB51
TSN (2 modalities) [13]	ImageNet	BN-Inception	94.0%	68.5%
P3D [17]	ImageNet	ResNet-50	88.6%	-
C3D [14]	Sports-1M	ResNet-18	85.8%	54.9%
ARTNet [35]	Kinetics	ResNet-18	94.3%	70.9%
TSM [5]	Kinetics	ResNet50	95.9%	70.7%
Ours	Kinetics	ResNet-50	**96.8%**	**73.4%**

**Table 7 sensors-22-06595-t007:** Accuracy and model parameters on Something-Something-V1 and Jester data sets.

Method	Param	Accuracy on Sth-Sth-V1	Accuracy on Jester
Baseline	23.68 M	45.6%	94.4%
+ME	24.53 M	46.0%	95.6%
+SAT	23.87 M	45.9%	95.1%
+ME + SAT	25.73 M	46.6%	96.2%
+ME + SAT + WS	25.73 M	**47.8%**	**96.6%**

**Table 8 sensors-22-06595-t008:** Ablation study on investigating 4 different layouts of integrating the 2 modules.

Combination	Mode 1	Mode 2	Mode 3	Mode 4
TOP-1	47.1	46.5	47.3	**47.8**

## Data Availability

Not applicable.

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
