# Peer review of "MEST: An Action Recognition Network with Motion Encoder and Spatio-Temporal Module"

_sensors, 2022, doi:10.3390/s22176595_

Round 1

Reviewer 1 Report

The authors present MEST, an efficient network to extract spatio-temporal information with relatively low computational load, which exhibits competitive performances than other popular methods. The results have demonstrated the effectiveness of our network in terms of accuracy, computational cost and

network scales.

1. The article should use the third person instead of "we".

2. English writting should be checked and improved throughout the manuscript.

3. References are not incomplete, especially 2022.

4. Experiments are not enough. Provide videos or expreriments to further verifies the contributions.

5. Writting format such as Eqs. (9) and (10) should be revised according to the template of journal.

6. The meanings of symbols should be provided after every equation.

Author Response

Reviewer # 1

  1. The article should use the third person instead of "we".

Response:

 Corrected as requested (in red)

  1. English writing should be checked and improved throughout the manuscript.

Response:

         We have proofread again, and made some grammar mistakes (in red)

  1. References are not incomplete, especially 2022.

Response:

         3 references are cited ([16], [21] and [33]) and described, on page 3 and 4 (in red)

  1. Experiments are not enough. Provide videos or experiments to further verifies the contributions.

Response:

         Action recognition has several directions:

  • Video based action recognition. Given a short video, the network aims to recognition the action in the video. In the training set, there are totally N types of actions, then for testing set, the network needs to calculate the possibilities for each action, and predict the action with the highest possibility value. For this category, the representative works include the following:
  • Temporal Segment Networks: Towards Good Practices for Deep Action Recognition
  • Two-Stream Convolutional Networks for Action Recognition in Videos
  • SlowFast Networks for Video Recognition
  • Skeleton based action recognition. The algorithms need to extract the skeleton of the human, and recognition its action based on it. Normally, they use OpenPose to estimate the pose and create a spatial-temporal network. The representative works include the following:
  • Spatial Temporal Graph Convolutional Networks for Skeleton-Based Action Recognition
  • Two-Stream Adaptive Graph Convolutional Networks for Skeleton-Based Action Recognition
  • Revisiting Skeleton-based Action Recognition
  • Temporal Action Localization (also called Temporal Action Detection). The video is trimmed into short snippets, which contain a certain types of action. The temporal action localization network needs to detect when the action takes place, and predicts the action type. The representative works include the following:
  • Background Suppression Network for Weakly-Supervised Temporal Action Localization
  • Learning to Refactor Action and Co-occurrence Features for Temporal Action Localization
  • Action Localization. Given a video, the network needs to locate the action using a bounding box. The representative works include the following:
  • You Only Watch Once: A Unified CNN Architecture for Real-Time Spatiotemporal Action Localization

Our network belongs to the first category. For video based action recognition, the typical experimental results are presented in the form of tables of recognition accuracy, confusion matrix and activation map. Unlike object detection or tracking, we cannot neither locate the action using a bounding box nor tell when the action happens.

  1. Writing format such as Eqs. (9) and (10) should be revised according to the template of journal.

Response:

         The template of Sensors does not specify the format of equations. We rewrite the equations using the math formula plug-in of Word processor.

  1. The meanings of symbols should be provided after every equation.

Response:

 Added as requested (in red)

Reviewer 2 Report

===== Synopsis:

The authors propose a network that classifies human motion units (actions). The study identifies weaknesses in other action recognition networks and build a leaner network that is based on one of the most popular DeepNets, the ResNet. The new system (MEST) shows better accuracy for most databases (than other networks), and in situations in which it does not, it has a much smaller weight base than others.

===== General Comments:

The study is well understandable. There is sufficient progress worth publishing. There are enough details given to reproduce the system. I am just wondering what the relation to autonomous driving is. If there does not exist a dataset for autonomous driving, then perhaps authors should make at least a relation to the topic in someway. Do those action collections not contain gestures that could be specifically discussed for the case of autonomous driving?, ie. classifying a person's gate.

===== Specific Comments:

- Citation [3] is from 2001. It must be an engineering study as the text suggest. But why do authors use the term CNN (conv neur. net)?

- Page 2, paragraph in the middle: ``Considering the fact that existing 2D CNN used for action recognition is still deep, ...'': sentence is confusing, it feels like it is missing some word.

Author Response

Reviewer # 2

I am just wondering what the relation to autonomous driving is. If there does not exist a dataset for autonomous driving, then perhaps authors should make at least a relation to the topic in someway. Do those action collections not contain gestures that could be specifically discussed for the case of autonomous driving?, ie. classifying a person's gate.

Response:

The manager has already transfer this paper to regular session.

===== Specific Comments:

- Citation [3] is from 2001. It must be an engineering study as the text suggest. But why do authors use the term CNN (conv neur. net)?

Response:

         We cited the wrong reference. It has been corrected on page 16 (in blue)

- Page 2, paragraph in the middle: ``Considering the fact that existing 2D CNN used for action recognition is still deep, ...'': sentence is confusing, it feels like it is missing some word.

Response:

         Modified as requested, on page 2 (in blue)